# Principal components of thermal regimes in mountain river networks

Daniel J. Isaak, Charles H. Luce, Gwynne L. Chandler, Dona L. Horan, Sherry P. Wollrab

U.S. Forest Service, Rocky Mountain Research Station, Aquatic Sciences Lab, Boise, ID 83702

*Correspondence to*: Daniel Isaak (disaak@fs.fed.us)

**Abstract.** Description of thermal regimes in flowing waters is key to understanding physical processes, enhancing predictive abilities, and improving bioassessments. Spatially and temporally sparse datasets, especially in logistically challenging mountain environments, have limited studies on thermal regimes but inexpensive sensors coupled with crowd-sourced data collection efforts provide efficient means of developing large datasets for robust analyses. Here, thermal regimes are assessed using annual monitoring records compiled from several natural resource agencies in the northwestern United States that spanned a five-year period (2011–2015) at 226 sites across several contiguous montane river networks. Regimes were summarized with 28 metrics and principal components analysis (PCA) was used to determine those metrics which best explained thermal variation on a reduced set of orthogonal axes. Four principal components (PC) accounted for 93.4% of the variation in the temperature metrics, with the first PC (49% of variance) associated with metrics that represented magnitude and variability and the second PC (29% of variance) associated with metrics representing the length and intensity of the winter season. Another variant of PCA, T-mode analysis, was applied to daily temperature values and revealed two distinct phases of spatial variability—a homogeneous phase during winter when daily temperatures at all sites were < 3 °C and a heterogeneous phase throughout the year's remainder when variation among sites was more pronounced. Phase transitions occurred in March and November, and coincided with the abatement and onset of subzero air temperatures across the study area. S-mode PCA was conducted on the same matrix of daily temperature values after transposition and indicated that two PCs accounted for 98% of the temporal variation among sites. The first S-mode PC was responsible for 96.7% of that variance and correlated with air temperature variation ($r = 0.92$) whereas the second PC accounted for 1.3% of residual variance and was correlated with discharge ($r = 0.84$). Thermal regimes in these mountain river networks were relatively simple and responded coherently to external forcing factors, so sparse monitoring arrays and small sets of summary metrics may be adequate for their description. PCA provided a computationally efficient means of extracting key information elements from the temperature dataset used here and could be applied broadly to facilitate comparisons among more diverse stream types and develop classification schemes for thermal regimes.

## 1 Introduction

Temperatures of flowing waters control many physicochemical processes (Likens and Likens, 1977; Gordon et al., 1991; Ducharne, 2007) and affect the ecology of aquatic organisms and communities (Isaak et al., 2017a; Neuheimer and Taggart, 2007; Woodward et al., 2010). Knowledge of thermal regimes, characterized as the annual sequence of temperature conditions specific to locations within river networks (Caissie, 2006), is key to understanding natural conditions and diagnosing anthropogenic impairments. Seminal work by Poff and colleagues (Poff and Ward, 1989; Poff et al., 1997) created a robust framework for describing flow regimes based on metric descriptions of magnitude, frequency, timing, duration, and variability that are largely transferrable to thermal

regimes (Poole et al., 2004; Olden and Naiman, 2010). Recent studies have contributed useful
derivations of temperature metrics (Arismendi et al., 2013; Chu et al., 2010; Rivers-Moore et al.,
2013; Steel et al., 2016) or classification schemes based on a small number of pre-selected metrics
(Maheu et al., 2016) but the limited availability of annual temperature records (Orr et al., 2015;
Isaak et al., 2018a) has slowed broad development and adoption of thermal regime concepts. Data
inadequacies are often compounded for montane riverscapes that are difficult to sample (Brown and
Hannah, 2008; Isaak et al., 2013), a shortfall that needs to be overcome given the importance of
these areas as climate refugia for cold-water biodiversity (Brown et al., 2009; Isaak et al., 2016a;
Quaglietta et al. 2018) and as the focus of costly regional conservation strategies (Roni et al., 2002;
Rieman et al., 2015).

Despite existing limitations, the importance of temperature to stream biota is well recognized and
inculcated to regulatory standards based on metrics used within threshold-based approaches (Poole
et al., 2004; Todd et al., 2008). Most often, those metrics represent some aspect of conditions during
warm summer months when temperature sensitive species or life stages are thought to be most
vulnerable (Ice et al., 2004; McCullough, 2010), which contributes to the preponderance of short
monitoring records spanning only these months (Isaak et al., 2017b). However, thermally mediated
ecological processes occur throughout the year (Neuheimer and Taggart, 2007; Olden and Naiman,
2010), so adequate understanding requires broader characterization of thermal conditions from
annual datasets. While that may bring additional complexity, most warm season metrics are strongly
correlated and therefore redundant (Isaak and Hubert, 2001; Dunham et al., 2005; Steel et al., 2016).
If redundancy is also the norm among a broader array of annual temperature metrics, then
multivariate data reduction techniques might be useful for identifying a few key aspects of thermal
regimes.

Supporting that idea, Rivers-Moore et al. (2013) used Principal Components Analysis (PCA) to
describe covariation among 39 temperature metrics calculated for 82 South African stream sites and
found that two PCs accounted for 75% of the total variation among metrics. Similarly in the field of
hydrology, Olden and Poff (2003) examined 171 flow metrics calculated from 420 gage sites across
the United States (U.S.) and found that two to four PCs accounted for 76–97% of variation in the
dataset. In addition to metric-based PCA that is commonly used in the hydrological sciences,
several other PCA variants are standard analytical tools in the field of climatology and may be
relevant for characterizing the dynamics of thermal regimes (Richman, 1986; Demsar et al., 2013).
Most notably, PCA can be done on repeated measurements of a single variable to identify common
spatial or temporal behavior among monitoring stations. In the climatology literature, for example,
empirical orthogonal function analysis (S-mode PCA in the taxonomy of Richman (1986)) is used
to determine which sites covary temporally as a means of developing regionalization schemes for
precipitation, air temperatures, or wind speeds (Piechota et al., 1997; Jimenez et al., 2008; Martins
et al., 2012). If common temporal patterns are identified, it suggests potential redundancy in the
monitoring network and the information can be used to refine future sampling designs. The closely
allied T-mode PCA identifies dominant spatial patterns in datasets and the times when these phases
occur (Richman, 1986; Gallacher et al., 2017). A single dominant spatial pattern suggests the spatial
distribution of a variable is temporally consistent whereas more than one spatial phase suggests
change points and different states.

The advent of inexpensive sensors, combined with regulatory requirements and concerns about
climate change, have led to the recent expansion in temperature monitoring networks for rivers and
streams (Isaak et al., 2010; Rivers-Moore et al., 2013; Hilderbrand et al., 2014; Luce et al., 2014a;
Trumbo et al., 2014; Hannah and Garner, 2015; Jackson et al., 2016; Molinero et al., 2015; Daigle
et al., 2016; Mauger et al., 2016; Steele et al., 2016). What was once a data dearth is becoming a
deluge and opportunities exist to study thermal regimes with robust datasets. Here, we use annual
temperature records compiled from several natural resource agencies for 226 monitoring sites in a
mountainous landscape to conduct an initial assessment of thermal regimes. We limit the
geographic scope of our effort to several adjacent river basins in the northwestern U.S. that are
geologically and topographically similar but which have particularly dense monitoring networks to
maximize analytical flexibility. Our objectives were to: 1) provide a basic description of the annual
thermal characteristics in mountain rivers and streams because these are rare within the literature, 2)
develop metrics to describe thermal regime characteristics based on magnitude, frequency, timing,
duration, and variability, and 3) explore spatiotemporal variation among those metrics and
temperature dynamics in relation to basin morphology and hydroclimatic conditions to better
discern the principal components of thermal regimes and their regulating factors.

**2 Study area**
The study area encompasses 79,500 km$^2$ of mountainous, topographically complex terrain that
spans a broad elevation range of 200–3,600 m at a latitude of 45° N (Figure 1). Climate is
characterized by cold, wet winters with moderate to heavy snow accumulations at high elevations
and hot, dry summers. Hydrographs are typical of snowmelt runoff systems, with high flows during
spring and early summer and low flows during late summer, fall, and winter (Figure 2). Vegetation
is dominated by conifer forests except at low elevations and south facing aspects where grasses and
shrubs predominate. Wildfires are common within the landscape and burned 8% of the area from
2011 to 2015 (Morgan et al., 2014). Parent geology consists mostly of resistant granites of the Idaho
Batholith and a smaller easterly portion of intrusive volcanics (Bond and Wood, 1978; Meyer et al.,
2001). Both geologies are heavily dissected and stream valleys are V-shaped except for some alpine
valleys at the highest elevations that were once glaciated. Human population densities are low
except along wider segments of river valleys where fertile floodplains and easy access to water
accommodate small amounts of agriculture and ranching. Most of the study area is publically
owned (81%) and federally administered by the United States National Forest Service and Bureau
of Land Management for a variety of land-use, recreational, and conservation purposes. Unpaved
road networks have been developed in some drainages for timber harvest but many drainages are
protected in large wilderness areas with minimal anthropogenic effects or roads (Swanson, 2015).

**2.1 River networks and temperature dataset**
Rivers and streams within the study area were delineated using the 1:100,000-scale National
Hydrography Dataset (NHD; http://www.horizon-systems.com/NHDPlus/index.php; McKay et al.,
2012), which was attributed with mean annual flow values from data at the Western U.S. Stream
Flow Metrics website
(http://www.fs.fed.us/rm/boise/AWAE/projects/modeled_stream_flow_metrics.shtml; Wenger et
al., 2010). To highlight the perennial subset of the network where temperature monitoring occurred,
reaches with annual flows less than 0.03 m$^3$/s were removed from the network, as were reaches with
channel slopes >15%, and those coded as intermittent in the NHD (Fcode = 46003). Filtering
reduced the original network extent from 58,000 km to 29,600 km with streams flowing at
elevations of 221–3,105 m. To visualize thermal heterogeneity in the network, a scenario
representing mean August temperatures for a baseline climate period of 1993–2011 was
downloaded from the Northwestern Stream Temperature website (NorWeST:
https://www.fs.fed.us/rm/boise/AWAE/projects/NorWeST.html; Isaak et al., 2016b) and linked to
the NHD reaches (Figure 1). Several large rivers drain the area in a generally westerly direction, the
largest of which is the Salmon River with a mean annual discharge of 315 m$^3$/s and a basin that
comprised 44% of the study area. Six large dams and reservoirs occur in downstream portions of the
network (three in the Boise River basin, two in the Payette River basin, and one in the Clearwater
River basin) but these affect thermal conditions in less than 300 km of river and no temperature data
were used from these sections. Spatial attributes and environmental characteristics of the study area
network are summarized in Table 1.

To obtain a water temperature dataset for analysis, we intersected the filtered network with the
NorWeST database of daily temperature summaries (Chandler et al., 2016) and extracted data for
sites that had mean daily temperature values on at least 70% of the days from December 1, 2010 to
November 30, 2015. We started the thermal year on December 1 because temperatures usually
reach their annual lows by this date and the 3-month period thereafter constituted a logical winter
season (i.e., December, January, February). Subsequent three-month periods were considered to be
spring (March, April, May), summer (June, July, August), and fall seasons (September, October,
November). NorWeST temperature records were supplemented with additional data solicited from
hydrologists and fisheries biologists employed by the Idaho Department of Fish and Game and the
U.S. Forest Service, and we also downloaded data from online databases maintained by the
Columbia Habitat Monitoring Program (https://www.champmonitoring.org/Home/Index) and the
NOAA Northwest Fisheries Science Center (https://www.webapps.nwfsc.noaa.gov/WaterQuality/).
Geographic gaps in monitoring were identified using geospatial analysis (e.g., Jackson et al., 2016)
and additional sensors were strategically deployed where needed (Isaak et al., 2010; 2013). Data
from the different sources were recorded at a variety of sub-daily intervals, so records were
summarized to mean daily temperatures for standardization. Data were collected using different
sensor models (TidbiT, Stowaway, and Pendant models from Onset Computer Corporation,
Pocasset, Massachusetts, USA; Temp101a model from MadgeTech, Warner, New Hampshire,
USA), which had measurement accuracies of +/-0.2°C to +/-0.5°C and resolutions of 0.02°C to
0.14°C based on manufacturer specifications and calibration tests we performed. Sensors were
deployed using underwater epoxy or steel cables for connection to large boulders and other
immobile channel structures and were shielded from direct sunlight (Isaak et al., 2013; Stamp et al.,
2014). Temperature records were subject to standard quality assurance-quality control measures as
described elsewhere (Chandler et al., 2016).

The stream temperature dataset consisted of records from 226 sites across a range of elevations,
stream sizes, and reach slopes (Figure 1; Table 1). Although we set the minimum threshold for
record completeness at 70% during the five-year period, the average completeness of records was
higher at 88%. Missing daily values were imputed using the MissMDA package (Missing Values
with Multivariate Data Analysis; Josse and Husson, 2016) in R (R Development Core Team, 2014)
because temporal covariation among proximate stream temperature sites is usually strong. That was
confirmed in our dataset by the high correlations between observed daily temperatures and
predictions from the imputation technique, which ranged from $r = 0.98$ to 0.99. All temperature
records at the 226 sites were complete after imputation and consisted of 1,826 mean daily
temperatures from December 1, 2010 to November 30, 2015. Climatological variation during the
same period was described using discharge data downloaded from the National Water Information
System database (https://waterdata.usgs.gov/usa/nwis/nwis) for a high-elevation gage site at 1,850
m and a low-elevation gage site at 294 m and air temperature data from monitoring stations in the
Cooperative Observer Network (https://www.ncdc.noaa.gov/data-access) that were near the gage
sites (Figure 1).

**3 Data analysis**
**3.1 PCA of thermal metrics**
Prior to calculating metrics for thermal characteristics, mean daily temperatures for 365 days were
calculated from the five years of data at each site to provide representative values. Twenty-eight
temperature metrics were then calculated to describe aspects of those annual records based on five
categories associated with magnitude, variability, frequency, timing, and duration (Tables 2 and 3).
Metrics were similar to those used in previous studies of thermal regimes (Arismendi et al., 2013;
Chu et al., 2010; Rivers-Moore et al., 2013; Steel et al., 2016) and in studies assessing the effects of
peak summer temperatures on the distribution and abundance of aquatic organisms (Dunham et al.,
2003; Huff et al., 2005; Isaak et al., 2017a). A wide range of variability occurred among sites where
mean annual temperatures ranged from 3.1 ℃ to 10.3 ℃ and annual standard deviations ranged
from 2.51 ℃ to 7.40 ℃ (Table 3). Relationships among the thermal metrics were described by
conducting PCA on a data matrix in which columns represented the 28 metrics and rows were the
226 monitoring sites. Linear combinations of the data were estimated with coefficients equal to the
eigenvectors of their correlation matrix, which were the principal components (PCs; Pearson, 1901;
Sergeant et al., 2016). The first principal component accounted for the largest possible variance in
the dataset and succeeding components accounted for the largest portions of the remaining variance
while being orthogonal (i.e., uncorrelated) to the preceding components. Correlations, or loadings,
between each metric and the PCs were also calculated to assist in subsequent interpretations. The
Princomp Procedure in SAS (SAS Institute Inc., 2015) was used to conduct the PCA. To describe
geographical relationships, PC scores were mapped to the 226 temperature sites and bivariate
correlations were calculated with descriptors of network conditions such as elevation, reach slope,
and discharge summarized in Table 1.

**3.2 PCA of daily water temperatures**
To assess the consistency of spatial temperature patterns among monitoring sites, a T-mode PCA
(Richman, 1986) was done on a data matrix of mean daily temperatures in which the columns were
the 365 days starting on December 1 and the rows were the 226 monitoring sites. In this analysis,
the number of principal components explaining significant variation indicates the number of distinct
spatial phases that occur throughout the year (Gallacher et al., 2016). Eigenvector loadings on the
dominant PCs were plotted for each day of the year to describe when each phase occurred, and
mean daily temperatures were mapped during these periods for visualization.

To assess temporal covariance among sites, an S-mode PCA (Richman, 1986) was done by
transposing the T-mode data matrix so that monitoring sites were columns and the time ordered
daily mean temperatures were rows. Because hydroclimatic conditions among years could have
affected the results, the S-mode PCA was done not only for the five-year averages of daily water
temperatures but also on the disaggregated time series of 1,826 daily values at the 226 monitoring
sites. Concordance between the S-mode PC scores, air temperature, and discharge were examined
posthoc by plotting standardized time-series and calculating bivariate correlations.

**4. Results**
Water temperatures within the study area network exhibited spatial and temporal variation that
reflected the local topography and annual hydroclimatic cycle. The annual temperature cycle is
illustrated in Figure 2 by the slopes of linear regressions between mean monthly temperatures and
elevation at the 226 monitoring sites throughout the course of the year in 2013. No elevation trend
occurred during cold winter months when many sites had water temperatures at or near 0˚C and
were frequently exposed to subzero air temperatures. As temperatures warmed during the spring a
small elevation trend appeared, which became most pronounced (approximately -0.37°C / 100 m)
during peak temperatures in the months of July and August. Examples of inter-annual variation are
shown in Figure 3, which contrasts the extreme conditions observed in 2011 and 2015. The former
year was relatively cool with a large winter snow accumulation and spring runoff, whereas 2015
had below average snowfall, low runoff, and particularly warm early summer air temperatures. As a
result, the median discharge date occurred 1–2 months earlier in 2015 than in 2011 and peak water
temperatures were 4–5 °C warmer.

Four PCs accounted for 93.4% of the variation in the 28 temperature metrics (Table 4). The first PC
explained 49% of the variation and was strongly correlated with metrics that represented magnitude
and variability during most seasonal periods. Correlations between PC1 scores and elevation ($r = -$
$0.59$) and mean flow ($r = 0.58$) suggested gradients in these network characteristics were important
controls on this component of thermal regimes (Table 5). PC2 explained 29% of thermal variation
and represented the length and intensity of the winter period, with strong loadings for mean winter
temperature, minimum temperature, and timing metrics that determined growing season length. PC3
accounted for 9.8% of total variation and was associated with summer temperature variability and
two timing metrics, whereas PC4 accounted for 5.6% of thermal variance. An ordination plot of
scores from the two dominant PCs showed a symmetrical distribution except for several sites with
large positive scores on the first axis that were from large rivers at low elevations and had the
warmest temperatures (Figure 4a). A map of PC1 scores indicated that the spatial pattern in
magnitude and variability (Figure 4b) was congruent with the network scenario of mean August
temperatures as would be expected (Figure 1). In fact, the correlation between PC1 scores and the
NorWeST August scenario predictions at the 226 monitoring sites was strong at $r = 0.86$. The PC2
map showed several clusters of stream sites with high scores scattered throughout the study area
(Figure 4c), which tended to be associated with lower reach slopes (Table 5).

In the T-mode analysis, the first two PCs explained 88% of the total variation in mean daily
temperatures. A plot of the daily eigenvector loadings indicated that one distinct spatial phase
occurred in the winter and a second phase spanned the year's remainder (Figure 5). Phase
transitions occurred around days 100 and 350, which closely aligned with the abatement and onset
of subzero air temperatures in the study area (Figure 2). Figure 6 illustrates the spatial patterns
characteristic of the two phases by mapping mean daily water temperatures at the monitoring sites
on days 50 and 250, which occurred in mid-January and late July, respectively. Temperatures
during the winter phase were spatially homogenous and exhibited a narrow range from 0 °C to 2.5
°C whereas the non-winter phase was heterogeneous and had a broader temperature range from 7.6
°C to 23.4 °C.

In the S-mode analysis, the first PC accounted for 98% of the variation when applied to the average
year of 365 daily temperatures at the 226 monitoring sites. Nearly an identical result was obtained
when the analysis was repeated on the disaggregated time-series of 1,826 daily temperatures, as
PC1 then explained 96.7% of total variation (Figure 7a). The correlation between PC1 scores and
mean daily air temperatures in the disaggregated series was strong ($r = 0.92$), suggesting that water
temperatures were responding coherently to variability in air temperatures across the study area. A
second PC accounted for 1.3% of water temperature variation in the disaggregated series and was
strongly correlated with variation in mean daily discharge ($r = 0.84$). A plot of PC1 versus PC2
indicated that variation along these axes corresponded to monthly and seasonal periods (Figure 7b).
As was expected, little variation occurred during the cold winter months but during spring and early
summer, variation was observed along both axes as air temperatures warmed and snowmelt runoff
created a large discharge pulse. Once discharge returned to baseflow conditions in late summer,
variability along PC1 was the primary signal until air temperatures cooled significantly in late fall
and the homothermous period began.

Although PC1 and PC2 are linearly uncorrelated, the loop structure of Figure 7b indicates there was
some mutual information and that one driver of temperature variation was out of phase with the
other. Examining this more closely by plotting the site loading values on each component from the
S-mode analysis, we see little variability among the loadings for PC1 relative to the much larger
range of loading values for PC2 (Figure 8). This confirms that PC1 represented the common
behavior among all stream sites and that deviations in timing of water temperature increases and
decreases were dictated by PC2. As a result, when annual temperature signals were reconstructed
for two sites from the PCs based on the mean loading value for PC1 and +/- 0.16 for PC2 to
represent strong negative and positive loadings, the expected timing shift was apparent (Figure 9).
Notably, the site with the -0.16 PC2 loading had a later, sharper rise in water temperature that
peaked in late summer approximately one month after the site with the positive loading. The
correspondence of PC2 with stream discharge in Figure 7a suggests the timing shift could be related
to runoff patterns. And indeed, the annual unit-area runoff for the basins associated with the 226
sites was a strong predictor of the PC2 loadings in a linear regression ($r^2 = 0.51$; Figure 10). Site
elevation provides some indication of rainfall-snowfall fraction that may help explain timing shifts
but this covariate added little predictive capacity beyond annual runoff when examined across all
sites ($r^2 = 0.54$). However, when sites with basin sizes less than 50 km$^2$ were examined (because
site elevation relates more strongly to mean basin elevation in smaller basins), elevation accounted
for a large increase in the explainable variance of PC2 loadings beyond that attributable to annual
runoff ($r^2 = 0.69$). Although orographic enhancement of precipitation is evident in the study area,
there is enough difference in circulation patterns across the north-south extent of the area that
elevation and annual runoff were only weakly correlated in the small basins ($r = -0.2$), so the
elevation effect was largely independent of annual precipitation. As a result, both factors appeared
to contribute to the PC2 loadings such that either wetter or colder locations had more negative
loadings and later rises in water temperatures.

**5 Discussion**
**5.1 Thermal regimes in mountain settings**
Thermal regimes in the mountain river networks we studied were simple and responded relatively
coherently to climatic variability across a geomorphically consistent area with few reservoirs.
Strong seasonal patterns in water temperatures characteristic of temperate latitudes were apparent in
response to the primary signal set by the annual air temperature cycle and accompanying changes in
solar radiation. Not surprisingly given the pronounced elevational gradients in the study landscape,
the dominant regime aspect represented by PC1 in the metric-based PCA was associated with
magnitude. Less expected was that many of the variability metrics also loaded heavily on the first
PC because variation has been treated as a distinct element of thermal regimes (e.g., Steel et al.,
2012; Kovach et al., 2018). The concurrence of magnitude and variability metrics probably also
relates to elevation and changes in the importance of groundwater buffering, which both cools
streams and dampens diurnal and seasonal variations (Cassie and Luce, 2017). For example, the
coldest streams at the highest elevations are usually strongly buffered by groundwater inputs
derived from large annual snowpacks in mountain environments and often show limited thermal
variability (Luce et al., 2014a; Isaak et al., 2016). Downstream from the headwaters, the
proportional inputs of groundwater decrease and streams are more coupled to climatic variability
even as their average temperatures increase due to solar gains over longer flow distances (Caissie
2006). In contrast to the metrics associated with PC1, metrics that described the winter period and
the extent of the growing season largely defined PC2. This "winter" PC is probably common to
stream thermal regimes in mountain landscapes where subzero air temperatures are frequent and
result in prolonged periods with water temperatures near 0 °C. The orthogonal nature of PC1 and
PC2 suggests that streams with otherwise similar magnitude and variance structures will sometimes
differ substantially with regards to their winter and growing seasons—a distinction that could have
important implications for biological communities or stream physicochemical processes.
Our results also suggest that important local nuances in water temperature dynamics like the
differences in timing of spring warming and peak temperatures may emerge from the interactions
among annual climate cycles, basin geomorphology, and hydrology. Because precipitation, air
temperatures, snowpack, runoff volume, and runoff timing are all evolving in response to climate
change in mountain environments across the study region (Mote et al., 2005; Luce et al., 2013) and
globally (Stewart, 2009), better understanding of these connections is needed. In particular, more
insight to the relationship of water temperatures with annual unit-area runoff and whether the
underlying mechanisms relate to changes in snowpack accumulation (Luce et al., 2014b; Lute and
Luce, 2017), snowmelt timing and rate (Musselman et al., 2017), the volume of water stored in
groundwater (e.g. Tague et al., 2007), or the outcomes of extreme low flows (e.g. Kormos et al.,
2016; Luce and Holden, 2009) could lead to better predictions about water temperatures and the
evolution of thermal regimes in response to expected changes in air temperatures and precipitation.
**5.2 Implications for modeling and monitoring**
Water temperature models are often developed for use in ecological assessments and to understand
how habitat degradation or restoration efforts may affect thermal regimes (Benyahya et al., 2007;
Gallice et al., 2015; Dugdale et al., 2017). Our results, like several previous studies that have
compared multiple temperature metrics (Isaak and Hubert, 2001; Rivers-Moore et al., 2013; Steele
et al., 2016), confirm that numerous metrics are strongly correlated and provide redundant
information. The specific choice of a metric, therefore, may not be critical as long as it represents an
important aspect of a thermal regime and is suited to the goals of a study. Metrics associated with
temperature magnitude and variability, which have been the focus of most modeling efforts, are
good choices because they represent significant portions of the information about thermal regimes
and have been shown on many occasions to be important determinants of ecological attributes such
as species distributions and abundance or physical processes in streams and rivers (Isaak et al.,
2017a; Webb et al., 2008). Our preferred metrics in previous research have been mean August or
mean summer temperatures because the data records for their calculation are typically available at
the largest number of sites in mountain environments, which maximizes sample sizes and
minimizes the distances over which interpolations are made when developing and applying
network-scale temperature models (e.g., Detenbeck et al., 2016; Isaak et al. 2017b). Metrics based
on longer-term means rather than short-term daily or weekly maxima are also more stable and easier
to predict (Isaak et al. 2010; Turschwell et al., 2016), although a focus on the latter metrics is often
mandated within regulatory environments and may negate these considerations (Todd et al., 2008;
McCullough 2010). Comparatively little effort has gone towards modeling temperature metrics
associated with growing season length or the dates of spring and winter season onset, despite the
significant information these metrics provide about thermal regimes and their relevance to the
phenology and life histories of organisms that constitute aquatic communities (Huryn and Wallace,
2000; Neuheimer and Taggart, 2007). These aspects of thermal regimes, as well as magnitude and
variability characteristics, are also likely to be evolving in response to climate change, so new
models are needed to provide forecasting abilities about changes later this century. Rather than
focusing on individual metrics, researchers could also instead use PCA to efficiently summarize
multiple temperature metrics and then model the eigenvector loadings that define one or more of the
principal components. This approach would maximize the amount of thermal information
represented by a response metric but would yield results that were more ambiguous to interpret.

The growth of new stream and river temperature monitoring and data collection activities has been
remarkable in recent years. Although optimization of those efforts ultimately depends on local
considerations, some general guidelines emerge from this work that may be applicable to other
areas. For example, the coherent behavior we observed among temperatures at many sites suggests
that a limited number of monitoring stations will often be sufficient to represent the temporal
dynamics of thermal regimes. Those stations would need to be spread geographically and along
major environmental gradients and replicated to mitigate against sensor losses, but 20–30 stations
might prove sufficient at scales comparable to our study area. Given low sensor costs and the
availability of standardized data collection protocols (Isaak et al., 2013; Stamp et al., 2014),
monitoring arrays could also be crowd-sourced effectively if site locations were coordinated and
chosen strategically using geospatial analyses to describe and stratify networks for sample
allocation (Jackson et al., 2016). Monitoring networks might also be supplemented by incorporating
data from sites established for other purposes such as documenting thermal responses to habitat
restoration efforts (Nichols and Ketcheson, 2013) or disturbances associated with land management,
wildfires, or livestock grazing (Mahlum et al., 2011; Nusslé et al., 2015). In fact, those factors
motivated collection of many of the datasets compiled for this analysis, although supplementation
with additional sites was needed to ensure adequate coverage within the study area.

If one of the goals of temperature data collection efforts is to develop accurate prediction maps that
show spatial variation in one or more thermal metrics (e.g., Isaak et al., 2017b; Steel et al., 2016),
monitoring sites may need to be established more densely than the temporal considerations
discussed above otherwise suggest. Spatial autocorrelation in temperature metric values is minimal
in mountain river networks beyond distances of 10–100 km (Isaak et al., 2010; Zimmerman and Ver
Hoef 2017), so this level of sensor spacing would be required to generate the most accurate maps.
Given the extent of many river networks, that could translate into a large number of sites but most
of these could be monitored for short periods while temporal dynamics were represented by a subset
of long-term sites because temporal covariance among sites would be strong. Costs associated with
numerous sensor deployments could be prohibitive, so aggregation of existing data sets from
multiple natural resource agencies into a centralized database often becomes an attractive option.
Moreover, if those central databases are made publically accessible, professionals from the
contributing agencies may begin to coordinate data collection activities more consistently and
effectively across larger areas (e.g., Isaak et al. 2018b).

As new data collection and database development efforts proceed, it is commonly the case that
temperature records have inconsistent period lengths or missing values. Usually it is desirable to
have complete records for analysis, so missing values are sometime imputed based on the
correlations between two monitoring site records that strongly covary (e.g., Rivers-Moore et al.,
2013). However, the process can be tedious if required at more than a few sites, so an efficient
improvement is offered by the imputation technique described by Josse and Husson (2012) that is
easily used in the MissMDA software package (Josse and Husson, 2016) for the R statistical
program (R Core Team, 2014). This technique examines and uses correlations among multiple site
records simultaneously to estimate missing values by first applying standard PCA to the incomplete
data set where missing values are replaced with the respective record mean. Data are then
reconstructed from the PCs, and the initial analysis step is repeated but with missing values replaced
using estimates from the reconstructed data. The process is repeated until convergence, and the
missing values in the original data records are ultimately replaced with estimates from the last PCA
reconstruction (Josse and Husson, 2012). Care should be taken against overreliance on the
technique to impute particularly sparse records but the MissMDA package provides a useful tool for
addressing gaps when working with large temperature datasets or time-series of other measurements
common to hydrology such as gage discharge records (e.g., Isaak et al., 2018a).

**5.3 Conclusions**
Our analysis of thermal regimes follows previous work that has proven fundamental to advancing
the understanding of hydrologic regimes (Poff et al., 1997; Olden and Poff, 2003) but also adds
novel applications of PCA variants from the field of climatology that hold utility for stream
temperature research and monitoring design. Insights from those analyses indicate that thermal
conditions in the mountain river networks studied here were strongly coherent through time,
exhibited two distinct spatial phases, could be adequately described by a few principal components
or allied metrics, and reflected landscape geomorphology and hydroclimatic conditions. A logical
next step involves application of these PCA techniques to larger stream and river temperature
datasets at regional, continental, or intercontinental scales to encompass greater heterogeneity and
discern the geographic domains over which distinct thermal regimes are operable. Across
sufficiently diverse landscapes, we might expect to find classes of thermal regimes that, at a
minimum, mimicked previously described classes of hydrologic regimes (e.g., rainfall, snowmelt,
spring-groundwater, and regulated) but possible divergences from, or additions to, those categories
would be useful to ascertain. In a national-scale assessment for the United States, Maheu et al.
(2015) classified stream thermal regimes into six categories but the 135 temperature stations that
supported the analysis were limited in comparison to a drainage network comprised of millions of
kilometers. Subsequent iterations on that effort could document additional, undescribed thermal
classes and might also prove beneficial by developing detailed maps of classification schemes to aid
in assessments of ecological conditions or anthropogenic effects on stream thermal regimes. As
research on the topic of thermal regimes matures, syntheses with flow regime concepts and
databases could also be sought to more fully describe the hydroclimatic conditions of flowing
waters.

*Data availability.* All water temperature data used in this study are available at the NorWeST website (https://www.fs.fed.us/rm/boise/AWAE/projects/NorWeST.html) whereas the full data set that includes air temperature and discharge data are available at the lead author's ResearchGate profile entry for this study (https://www.researchgate.net/profile/Daniel_Isaak).

*Competing interests.* The authors declare that they have no conflict of interest.

*Acknowledgements.* We thank Dave Schoen, Bart Gamett, Dan Garcia, Scott Vuono, Caleb Zurstadt, and Clayton Nalder with the U.S. Forest Service, Tim Copeland, Eric Stark, and Ron Roberts with the Idaho Department of Fish and Game, Eric Archer and Jeff Ojala with the Pacfish-Infish Biological Opinion monitoring program, and Boyd Bowes and Chris Jordan with the CHaMP monitoring program that contributed water temperature data to enable this research. Comments from Nicholas Rivers-Moore and one anonymous reviewer improved the quality of the final manuscript. The authors of this work were supported by the U.S. Forest Service Rocky Mountain Research Station.

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

**Table 1.** Descriptive statistics for spatial attributes of the study network and 226 monitoring sites
with annual temperature data in the northwestern United States.

| Network reaches | Mean | Median | SD | Minimum | Maximum |
|---|---|---|---|---|---|
| Elevation (m) | 1,493 | 1,533 | 536 | 221 | 3,105 |
| Drainage area (km$^2$) | 915 | 17.7 | 4,359 | 0.005 | 34,865 |
| Mean annual flow (m$^3$/s) | 9.73 | 0.229 | 43.2 | 0.0253 | 379 |
| Reach slope (m/m) | 0.0584 | 0.0519 | 0.0429 | 0 | 0.150 |
| | | | | | |
| Monitoring sites | | | | | |
| Elevation (m) | 1,392 | 1,407 | 464 | 280 | 2,369 |
| Drainage area (km$^2$) | 687 | 47.3 | 3,011 | 2.18 | 34,865 |
| Mean annual flow (m$^3$/s) | 7.37 | 0.692 | 26.4 | 0.0253 | 281 |
| Reach slope (m/m) | 0.0389 | 0.0273 | 0.0403 | 0 | 0.150 |


**Table 2.** Temperature metrics used to describe thermal regimes of mountain rivers and streams.

| Category | Thermal metric | Definition |
|---|---|---|
| Magnitude | M1. Mean annual temperature | Average of mean daily temperatures during a year |
| | M2. Mean winter temperature | Average of mean daily temperatures during December, January, and February |
| | M3. Mean spring temperature | Average of mean daily temperatures during March, April, and May |
| | M4. Mean summer temperature | Average of mean daily temperatures during June, July, and August |
| | M5. Mean August temperature | Average of mean daily temperatures during August |
| | M6. Mean fall temperature | Average of mean daily temperatures during September, October, and November |
| | M7. Minimum daily temperature | Lowest mean daily temperature during a year |
| | M8. Minimum weekly average temperature | Lowest seven-day running average of mean daily temperature during a year |
| | M9. Maximum daily temperature | Highest mean daily temperature during a year |
| | M10. Maximum weekly average temperature | Highest seven-day running average of mean daily temperature during a year |
| | M11. Annual degree days | Cumulative total of degree days during a year (1°C for 24 hours = 1 degree day) |
| Variability | V1. Annual standard deviation | Standard deviation of mean daily temperature during a year |
| | V2. Winter standard deviation | Standard deviation of mean daily temperature during winter months |
| | V3. Spring standard deviation | Standard deviation of mean daily temperature during spring months |
| | V4. Summer standard deviation | Standard deviation of mean daily temperature during summer months |
| | V5. Fall standard deviation | Standard deviation of mean daily temperature during fall months |
| | V6. Range in extreme daily temperatures | Difference between minimum and maximum mean daily temperatures during a year (M9 minus M7) |
| | V7. Range in extreme weekly temperatures | Difference between minimum and maximum weekly average temperatures during a year (M10 minus M8) |
| Frequency | F1. Frequency of hot days | Number of days with mean daily temperatures >20 °C |
| | F2. Frequency of cold days | Number of days with mean daily temperatures <2 °C |
| Timing | T1. Date of 5% of degree days | Number of days from December 1st until 5% of degree days are accumulated |
| | T2. Date of 25% of degree days | Number of days from December 1st until 25% of degree days are accumulated |
| | T3. Date of 50% of degree days | Number of days from December 1st until 50% of degree days are accumulated |
| | T4. Date of 75% of degree days | Number of days from December 1st until 75% of degree days are accumulated |
| | T5. Date of 95% of degree days | Number of days from December 1st until 95% of degree days are accumulated |
| Duration | D1. Growing season length | Number of days between the 95% and 5% of degree days (T5 minus T1) |
| | D2. Duration of hot days | Longest number of consecutive days with mean daily temperatures >20 °C |
| | D3. Duration of cold days | Longest number of consecutive days with mean daily temperatures <2 °C |


**Table 3.** Descriptive statistics for temperature metrics used to describe thermal regimes at 226
monitoring sites in mountain river networks. Statistics were calculated from the imputed time-series
and mean daily values for the period 2011–2015.

| | Mean (℃) | Median (℃) | SD (℃) | Minimum (℃) | Maximum (℃) |
|---|---|---|---|---|---|
| M1. Mean annual temperature | 5.36 | 5.10 | 1.44 | 3.10 | 10.34 |
| M2. Mean winter temperature | 0.75 | 0.63 | 0.60 | -0.10 | 4.03 |
| M3. Mean spring temperature | 3.67 | 3.47 | 1.61 | 1.14 | 9.38 |
| M4. Mean summer temperature | 11.2 | 10.9 | 2.68 | 6.55 | 19.1 |
| M5. Mean August temperature | 12.5 | 12.1 | 2.78 | 7.78 | 22.5 |
| M6. Mean fall temperature | 5.71 | 5.50 | 1.53 | 3.04 | 11.5 |
| M7. Minimum daily temperature | 0.21 | 0.14 | 0.35 | -0.45 | 2.18 |
| M8. Minimum weekly average temperature | 0.31 | 0.23 | 0.40 | -0.42 | 2.69 |
| M9. Maximum daily temperature | 13.5 | 13.0 | 3.00 | 8.26 | 23.5 |
| M10. Maximum weekly average temperature | 13.2 | 12.7 | 2.99 | 7.96 | 23.2 |
| M11. Annual degree days | 1956 | 1863 | 527 | 1132 | 3775 |
| V1. Annual standard deviation | 4.43 | 4.27 | 1.05 | 2.51 | 7.40 |
| V2. Winter standard deviation | 0.30 | 0.29 | 0.16 | 0.00 | 0.87 |
| V3. Spring standard deviation | 1.62 | 1.57 | 0.72 | 0.33 | 5.36 |
| V4. Summer standard deviation | 1.99 | 1.88 | 0.61 | 0.61 | 4.45 |
| V5. Fall standard deviation | 3.43 | 3.34 | 0.73 | 2.13 | 6.05 |
| V6. Range in extreme daily temperatures | 13.3 | 12.8 | 3.06 | 7.50 | 23.3 |
| V7. Range in extreme weekly temperatures | 12.9 | 12.3 | 3.06 | 6.99 | 22.9 |
| F1. Frequency of hot days | 0.81 | 0 | 5.82 | 0 | 61 |
| F2. Frequency of cold days | 131 | 132 | 35.6 | 0 | 212 |
| T1. Date of 5% of degree days | 109 | 113 | 25.5 | 44 | 168 |
| T2. Date of 25% of degree days | 193 | 194 | 10.9 | 148 | 217 |
| T3. Date of 50% of degree days | 237 | 238 | 5.01 | 215 | 251 |
| T4. Date of 75% of degree days | 276 | 276 | 2.99 | 264 | 288 |
| T5. Date of 95% of degree days | 323 | 323 | 4.78 | 309 | 340 |
| D1. Growing season length | 214 | 210 | 29.7 | 141 | 296 |
| D2. Duration of hot days | 0.691 | 0 | 5.61 | 0 | 61 |
| D3. Duration of cold days | 124 | 124 | 39.0 | 0 | 207 |


**Table 4.** Loadings of 28 temperature metrics on the first four principal components in a PCA of
annual temperature records from mountain river networks in the northwestern United States.

| Temperature metric | PC1 | PC2 | PC3 | PC4 |
|---|---|---|---|---|
| M1. Mean annual temperature | 0.99 | -0.07 | -0.05 | -0.03 |
| M2. Mean winter temperature | 0.26 | -0.92 | 0.14 | 0.00 |
| M3. Mean spring temperature | 0.91 | -0.19 | -0.25 | 0.04 |
| M4. Mean summer temperature | 0.97 | 0.21 | -0.06 | -0.05 |
| M5. Mean August temperature | 0.95 | 0.22 | 0.16 | -0.10 |
| M6. Mean fall temperature | 0.96 | -0.18 | 0.14 | -0.08 |
| M7. Minimum daily temperature | -0.02 | -0.86 | 0.08 | -0.02 |
| M8. Minimum weekly average temperature | -0.03 | -0.90 | 0.08 | 0.00 |
| M9. Maximum daily temperature | 0.95 | 0.26 | 0.09 | -0.08 |
| M10. Maximum weekly average temperature | 0.95 | 0.25 | 0.09 | -0.07 |
| M11. Annual degree days | 0.99 | -0.07 | -0.05 | -0.03 |
| V1. Annual standard deviation | 0.90 | 0.41 | 0.01 | -0.07 |
| V2. Winter standard deviation | 0.69 | -0.54 | 0.16 | 0.00 |
| V3. Spring standard deviation | 0.71 | 0.30 | -0.55 | 0.04 |
| V4. Summer standard deviation | 0.42 | 0.32 | 0.78 | -0.14 |
| V5. Fall standard deviation | 0.87 | 0.39 | 0.19 | -0.12 |
| V6. Range in extreme daily temperatures | 0.93 | 0.33 | 0.08 | -0.07 |
| V7. Range in extreme weekly temperatures | 0.93 | 0.33 | 0.08 | -0.07 |
| F1. Frequency of hot days | 0.47 | -0.01 | 0.30 | 0.82 |
| F2. Frequency of cold days | -0.70 | 0.61 | 0.09 | 0.11 |
| T1. Date of 5% of degree days | 0.02 | 0.96 | -0.10 | 0.01 |
| T2. Date of 25% of degree days | -0.43 | 0.74 | 0.46 | -0.08 |
| T3. Date of 50% of degree days | -0.45 | 0.37 | 0.79 | -0.16 |
| T4. Date of 75% of degree days | -0.19 | -0.51 | 0.72 | -0.19 |
| T5. Date of 95% of degree days | 0.30 | -0.88 | 0.12 | -0.09 |
| D1. Growing season length | 0.03 | -0.97 | 0.11 | -0.03 |
| D2. Duration of hot days | 0.44 | -0.03 | 0.32 | 0.84 |
| D3. Duration of cold days | -0.64 | 0.66 | 0.07 | 0.11 |
| Variance explained (%): | 49.0% | 29.0% | 9.8% | 5.6% |
| Cumulative variance (%): | 49.0% | 78.0% | 87.8% | 93.4% |
| Eigenvalue: | 13.73 | 8.12 | 2.74 | 1.56 |


**Table 5.** Correlations among stream temperature principal components and spatial attributes of 226
monitoring sites with annual data from river networks in the northwestern United States.

| | Elevation | Mean flow | Reach slope | PC1 | PC2 | PC3 | PC4 |
|---|---|---|---|---|---|---|---|
| Elevation | 1 | | | | | | |
| Mean flow | -0.34 | 1 | | | | | |
| Reach slope | -0.10 | -0.23 | 1 | | | | |
| PC1 | -0.59 | 0.58 | -0.34 | 1 | | | |
| PC2 | 0.27 | -0.06 | -0.49 | 0.00 | 1 | | |
| PC3 | -0.23 | 0.35 | 0.13 | 0.00 | 0.00 | 1 | |
| PC4 | 0.12 | 0.54 | -0.02 | 0.00 | 0.00 | 0.00 | 1 |


**Fig. 1.** Locations of 226 monitoring sites overlaid on an August stream temperature scenario for the 29,600 km network in the study area. Stars denote where air temperature and stream discharge data were obtained from a low-elevation site (294 m, northern station) and a high-elevation site (1850 m, southern station).

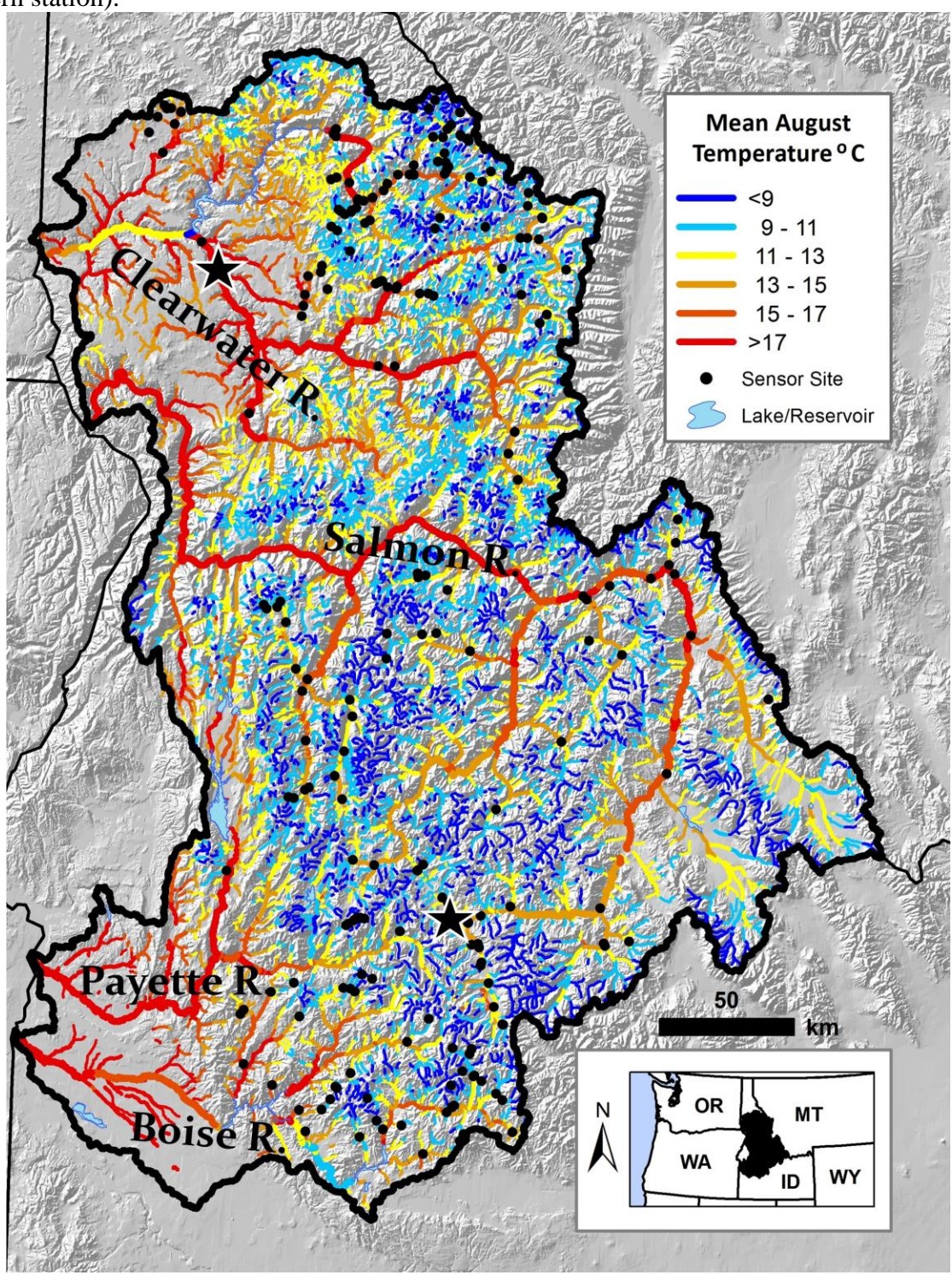

**Fig. 2.** Linear regression trends between elevation and mean monthly temperatures at 226 river and
stream sites during 2013 (data values are not shown for clarity). Values next to the trend lines are
regression slopes and $r^2$ values from the regressions.

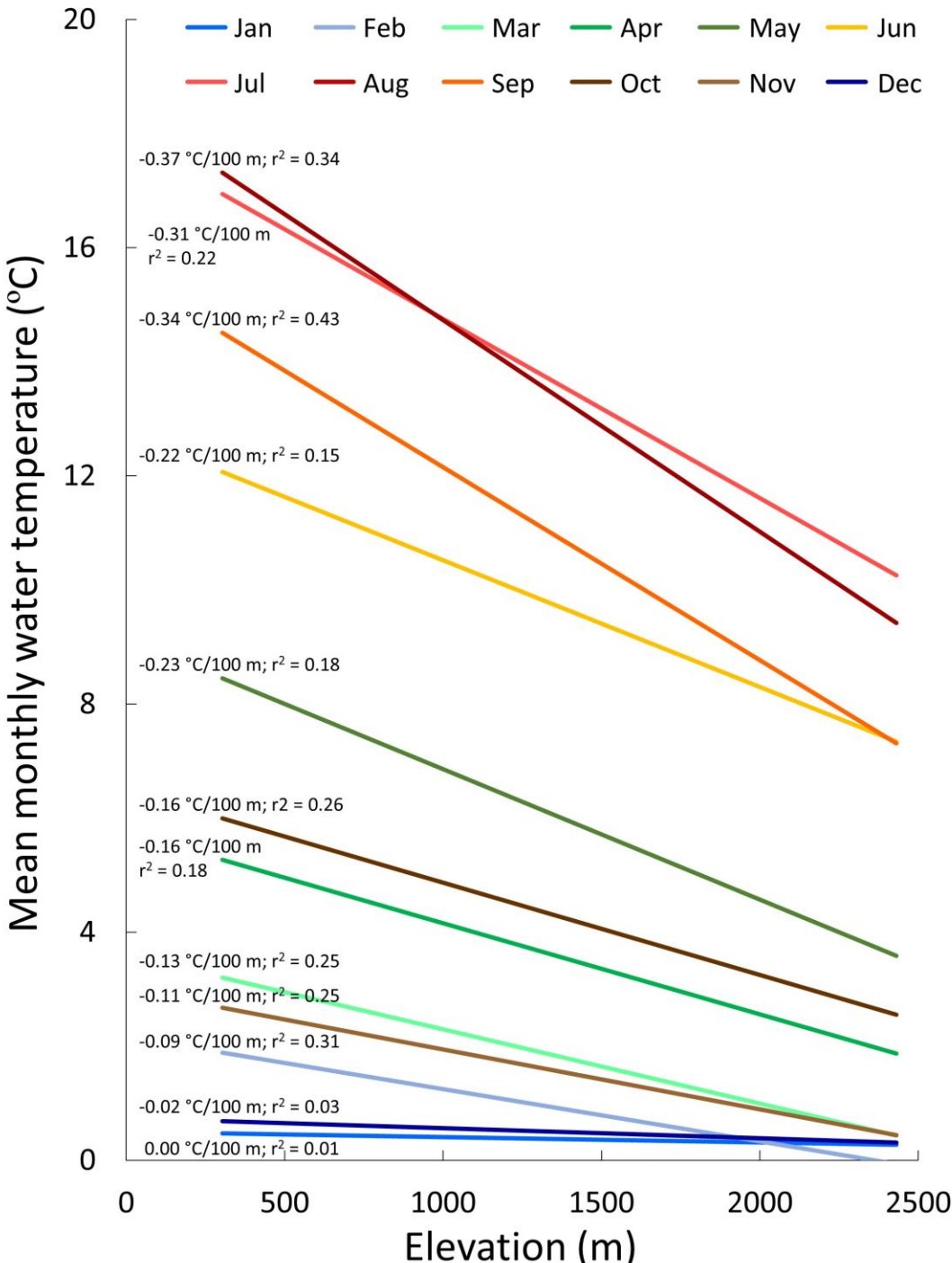


**Fig. 3.** Annual cycle of mean daily water temperatures (a), air temperatures (b), and discharge (c) at
a high-elevation site and a low-elevation site during two contrasting climate years. Discharge values
at the high elevation site are multiplied by ten for better visibility.

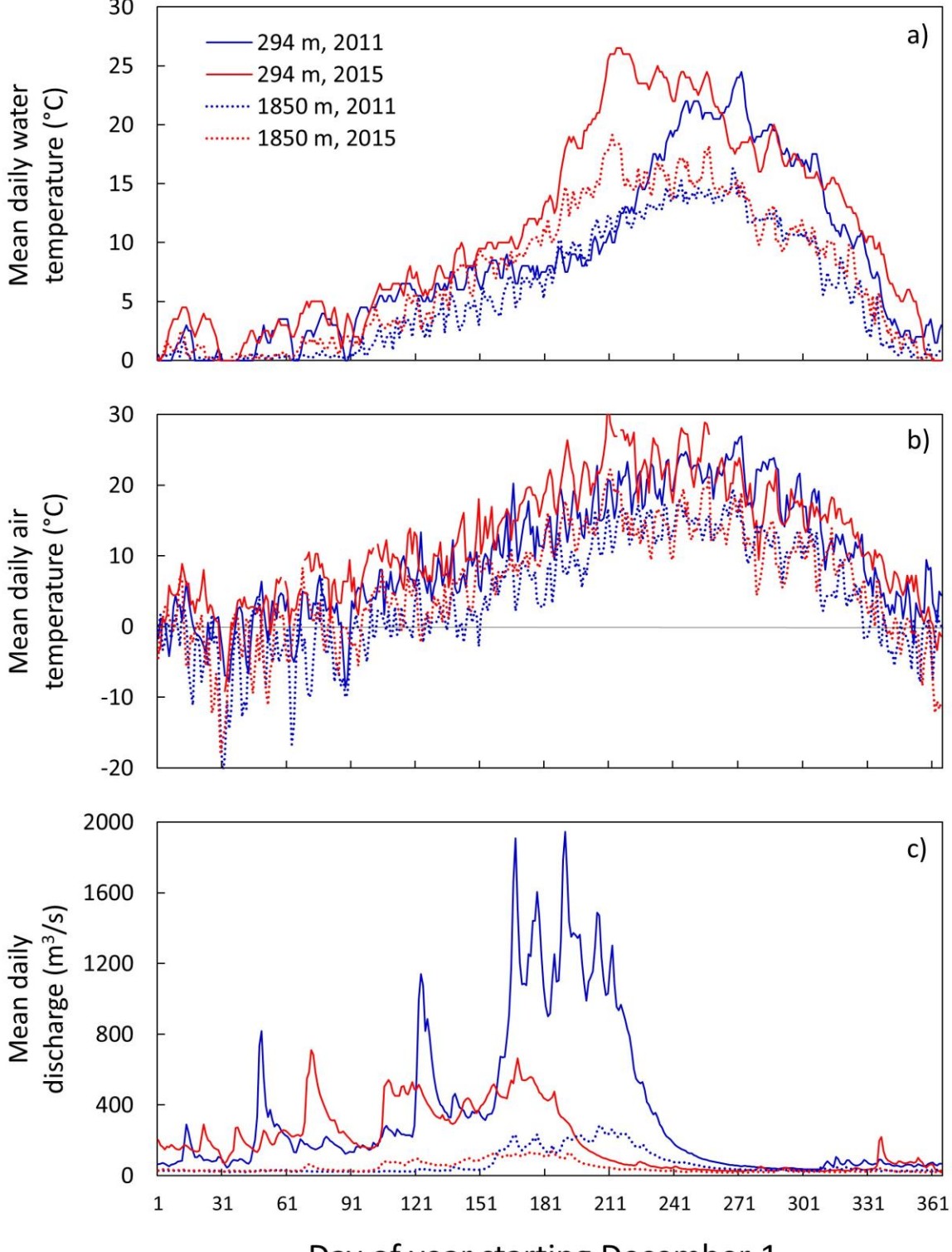


**Fig. 4.** Ordination plot that shows principal component scores of the first two axes derived from
water temperature data measured at 226 sites and summarized with 28 thermal metrics (a). Panels b
and c show principal component scores mapped to network locations.

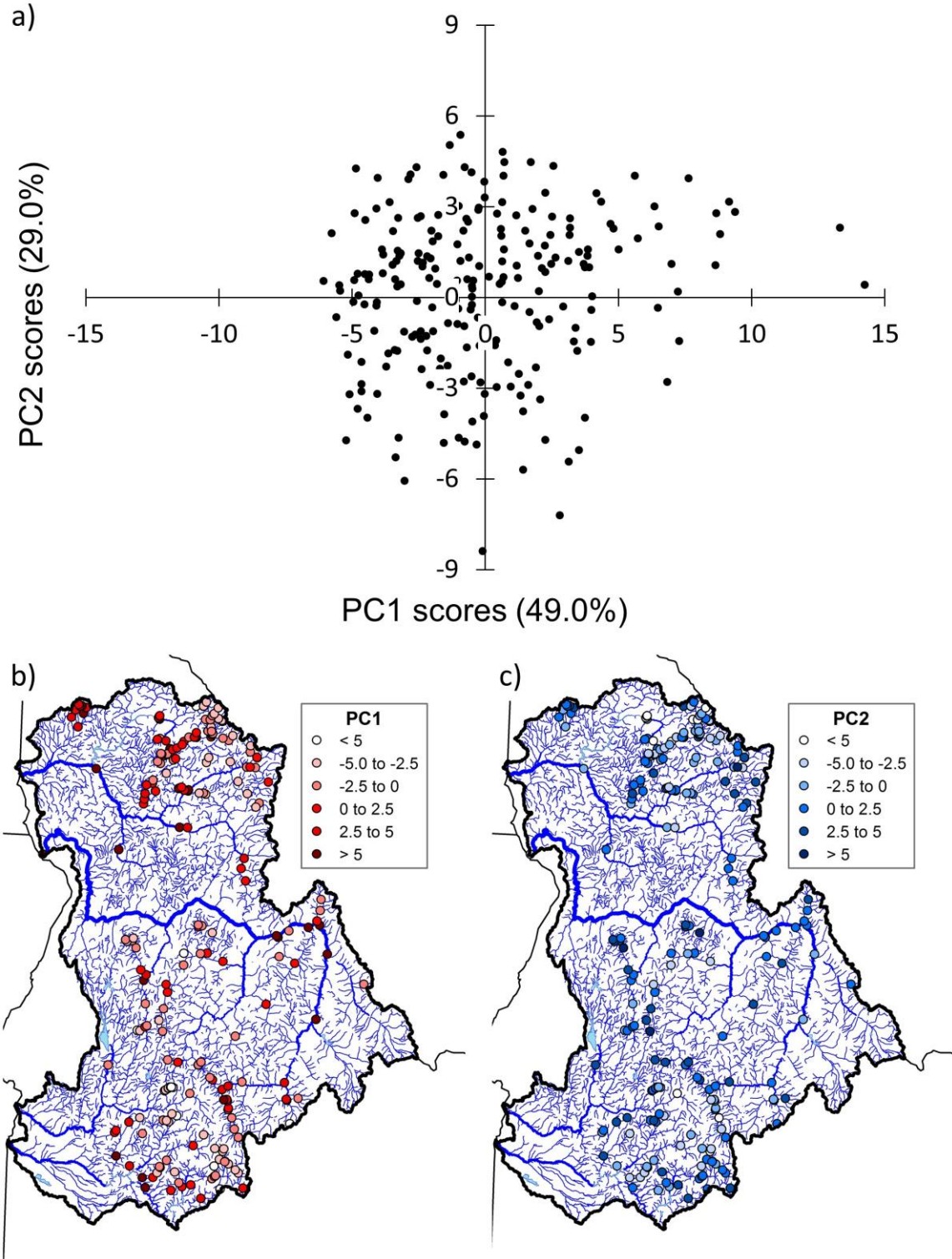

**Fig. 5.** T-mode PCA results showing times when dominant spatial phases occurred in water
temperatures at 226 sites based on principal component eigenvector loadings during an average
year.

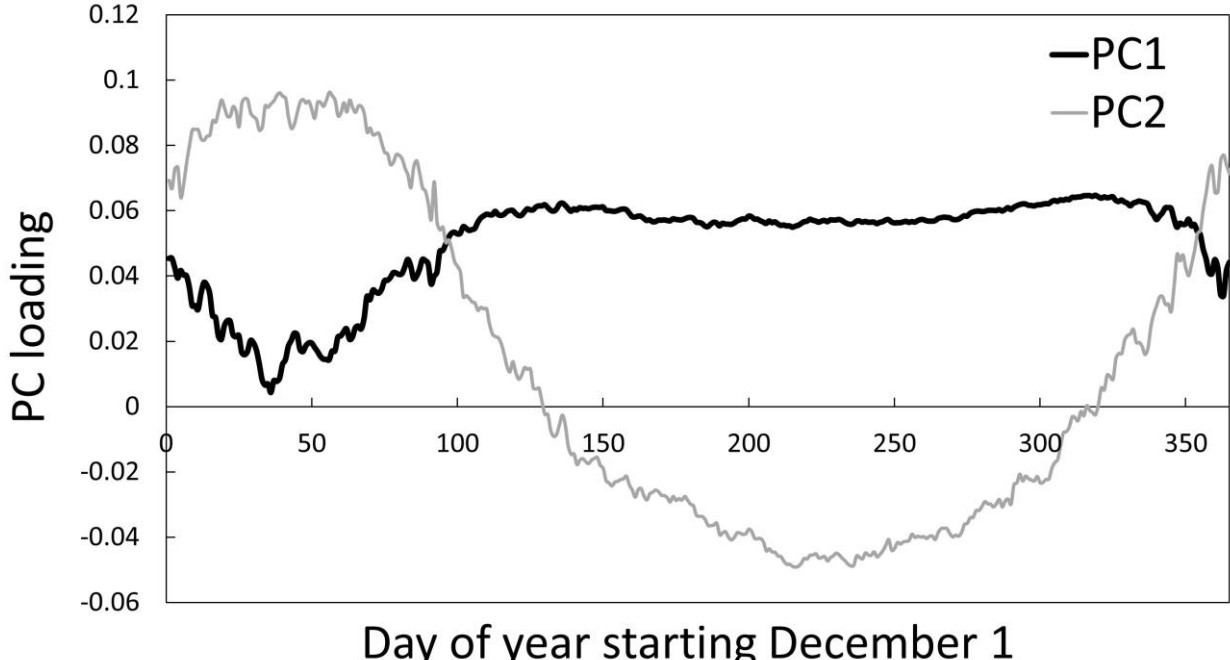


 **Figure 6.** Thermal patterns during two periods with distinct spatial phases based on T-mode PCA
results (a). Day 50 occurs in mid-January and represents the homogenous winter period (b) whereas
day 250 occurs in late July and represents the heterogeneous period (c).

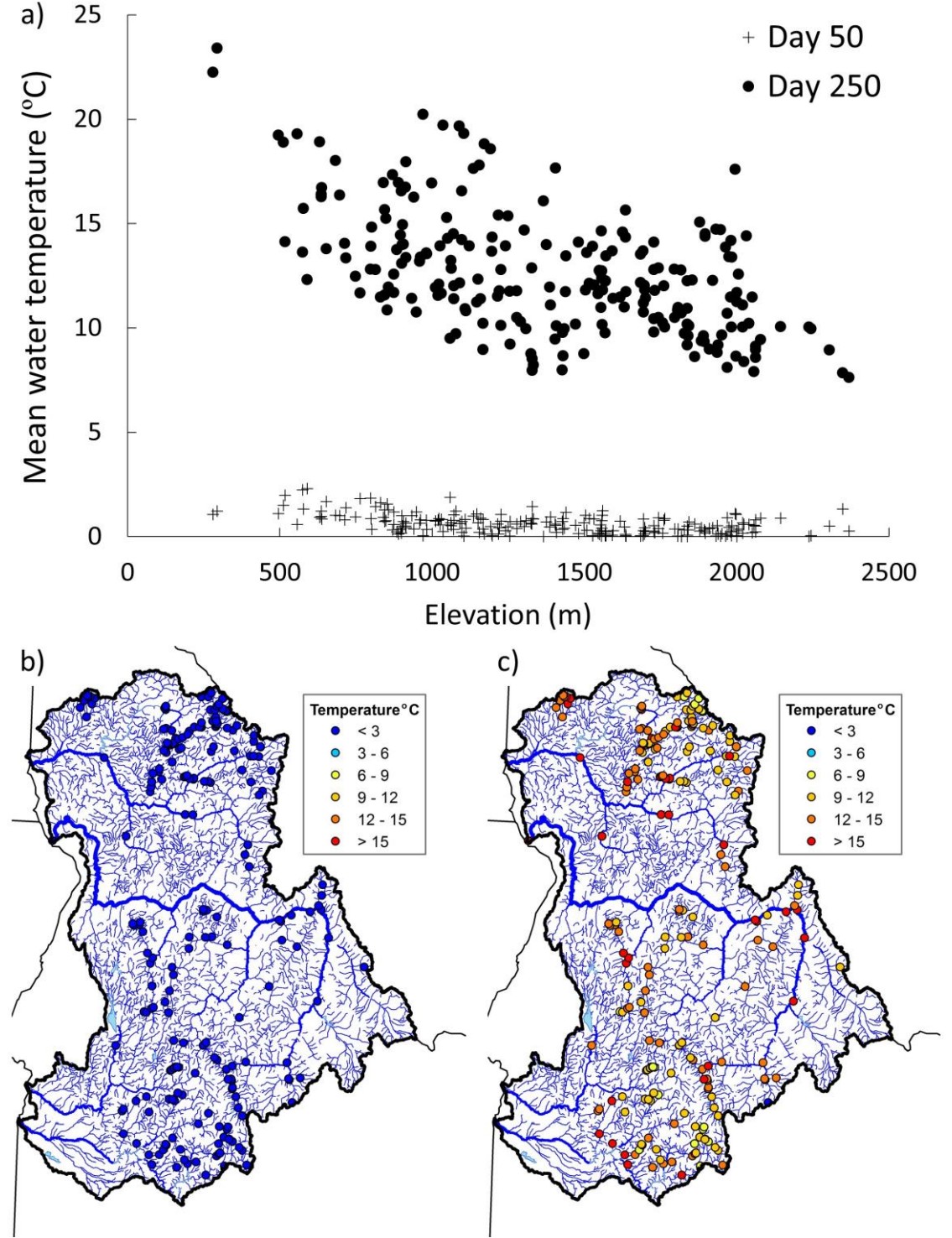

**Fig. 7.** S-mode PCA results showing principal component scores that describe temporal patterns in
mean daily water temperatures for 226 stream sites during five years (a). Average daily air
temperatures and discharge values from two monitoring stations are aligned with the principal
component scores for comparative purposes. A plot of PC1 versus PC2 reveals that variation along
the two axes differs by monthly and seasonal periods (b).

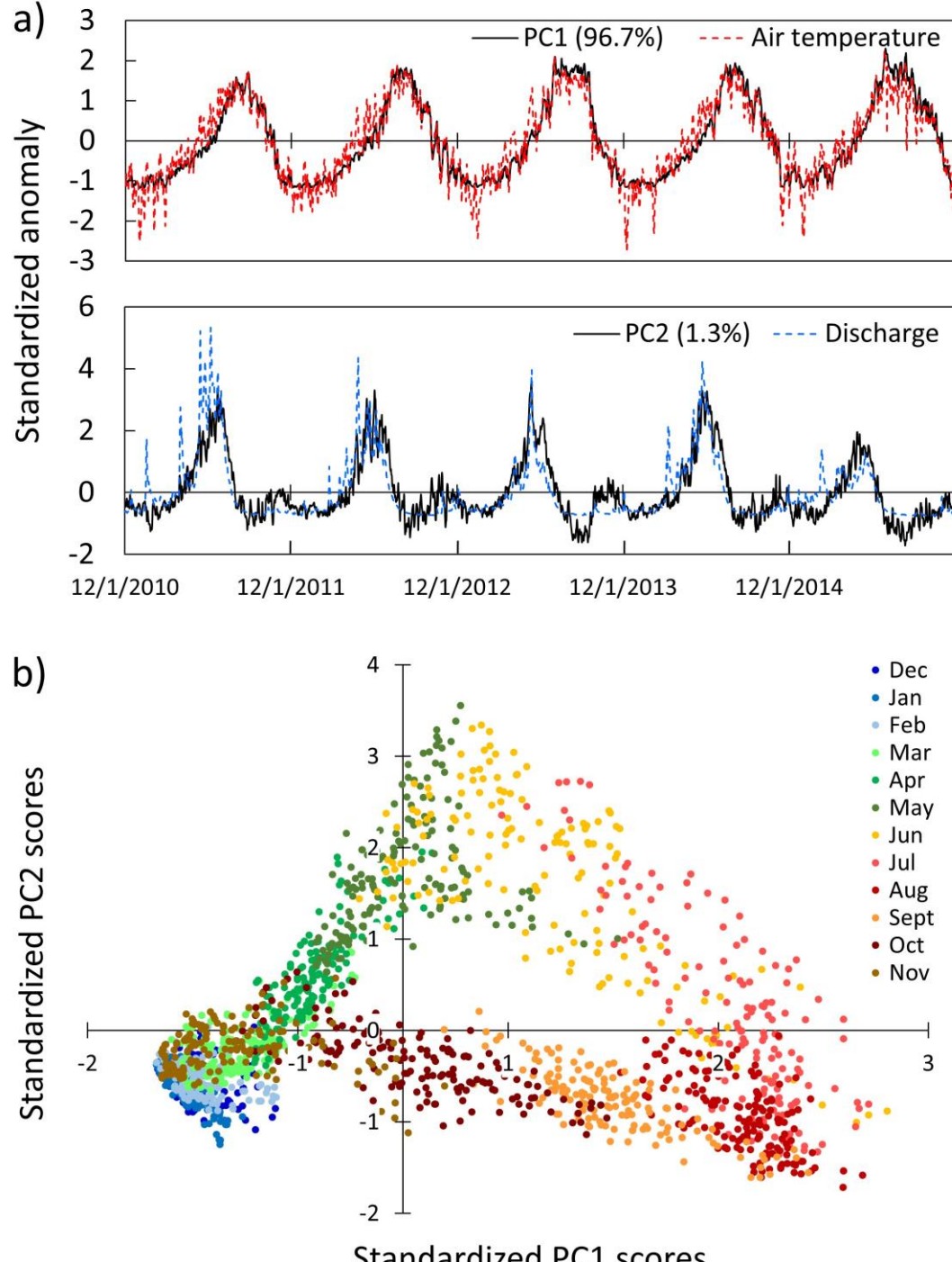


**Fig. 8.** Plot of S-mode eigenvector loadings from 226 stream sites on PC1 and PC2. Note that the
range of variation in the PC1 loadings is small relative to the loadings along PC2, which indicates
that most of the differences among sites were associated with the second principal component.

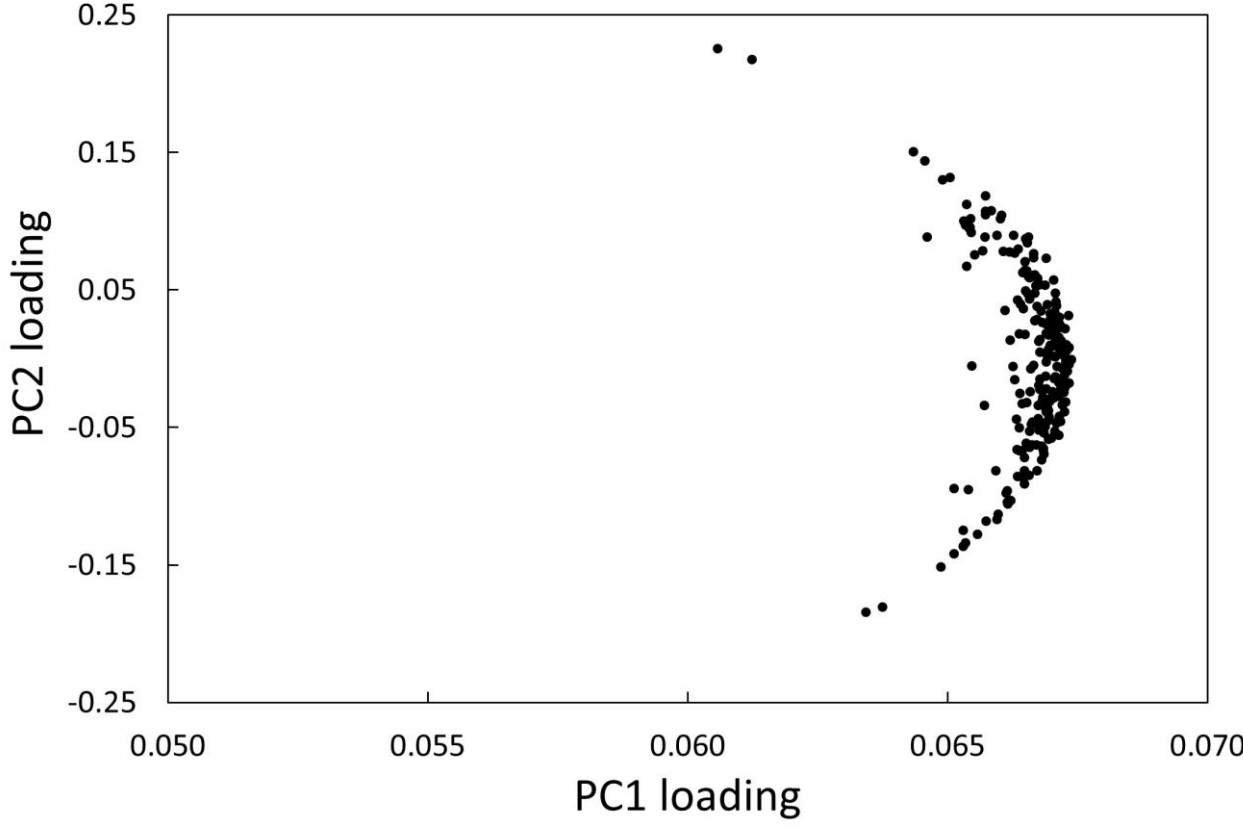


**Fig. 9.** Annual water temperature timing patterns reconstructed from S-mode PCs using the mean
eigenvector loading value for PC1 and +/- 0.16 for PC2 to demonstrate the effects of strong
negative loadings and positive loadings on PC2.

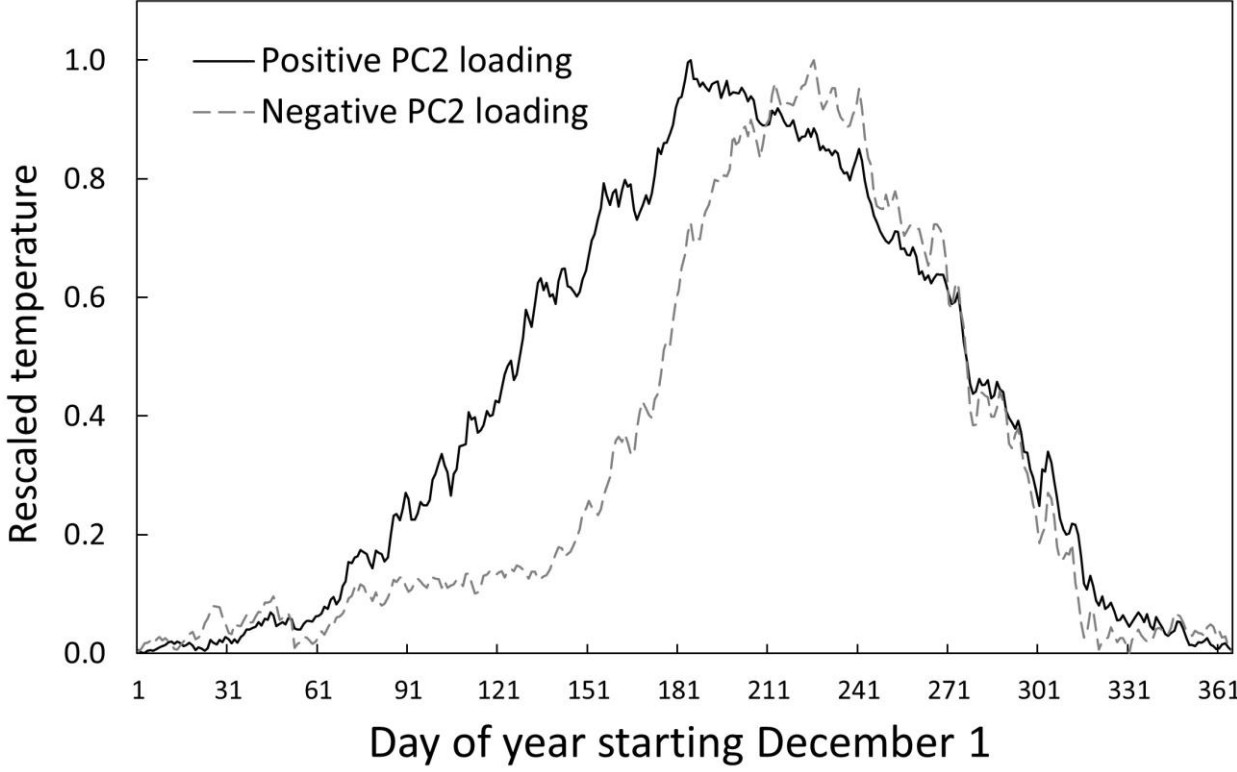


**Fig. 10.** Relationship between the S-mode eigenvector loadings from PC2 and the annual unit-area
runoff in basins upstream of 226 water temperature sites.

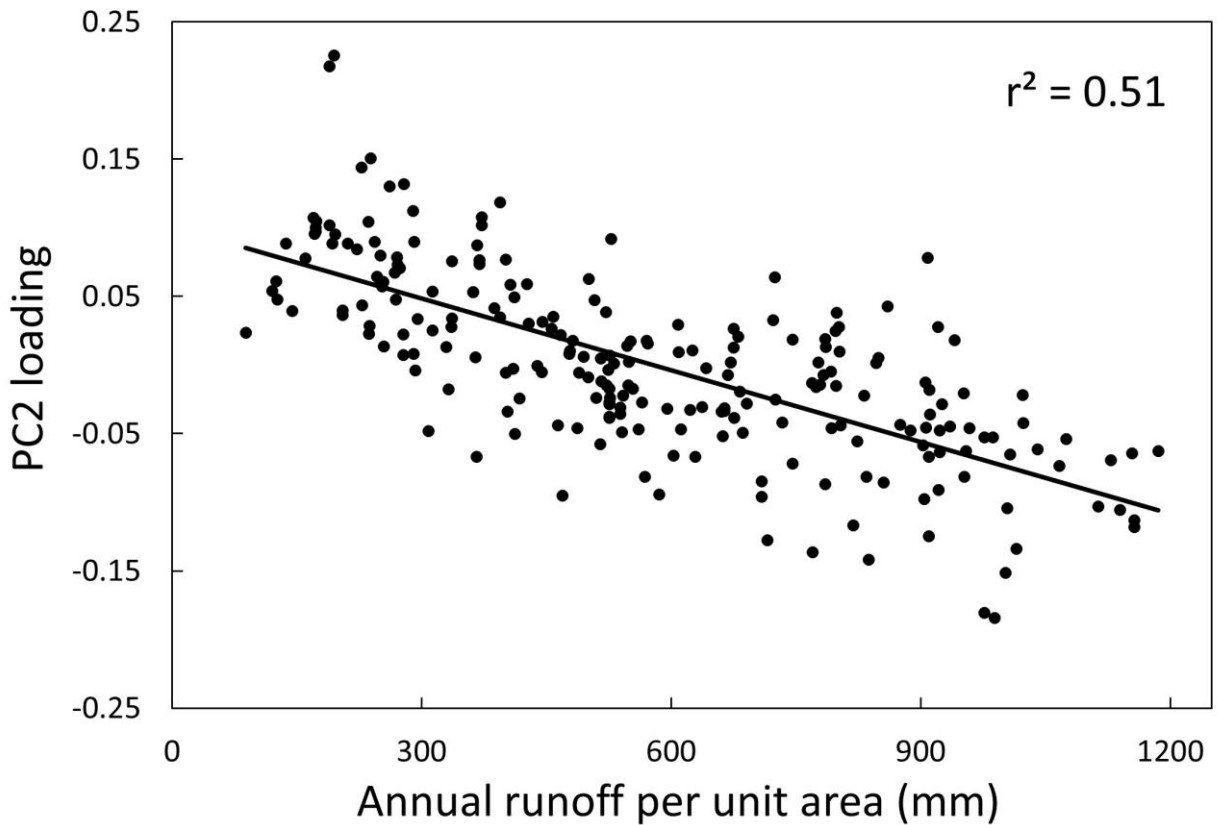
