# Peer review of "Principal components of thermal regimes in mountain river networks"

_Hydrology and Earth System Sciences, 2018_

## Referee Comment (RC1) · N. Rivers-Moore (Referee) · 7 Aug 2018

Specific comments: Lines 37-40: Sentence does not read well. Suggested revision "Knowledge of the local thermal regime, based on the annual sequence of temperatures characteristic to specific locations within a river network, is key to understanding natural conditions and diagnosing anthropogenic impairments." Lines 62-64: Suggested revision "While that may bring..., most warm stream...correlated with each other and therefore redundant. If redundancy is also reflected across a broader..." Paragraph beginning line 146: Be explicit that these time series refer to water temperatures, as later on in the manuscript air temperatures are also used. Section 3.1 – Does one need to specify that the study assumed stationarity in the data, in order to generate temperatures for 365 days based on five-year time series? Line 207 – Please provide

a summary of the environmental gradients; it may be worth including a table on these. Line 212 – Please explain why the thermal year started on 1 December. In South Africa, we typically use 1 October – 30 September for the Hydrological year, but I am aware that this varies regionally, being based on the onset of the highest discharge season. Line 218 – "sites, an S-mode" Line 234 – It would make more sense to me to represent the thermal gradient per 100m. This would be a useful figure in defining a water temperature lapse rate. For air temperatures, this is typically expressed as something like 0.7°C per 100m. Line 253 – Figure 4a Line 257 – insert Figure 4c Table 1 – write US in full; standardise on number of decimal points down columns (also applies for Table 3). Figure 2 – I like this figure! Please include the range of R2 values, and I would recommend that the caption explicitly describes the month(s) with the highest thermal gradient. Figure 4 – caption revision to say "...show principal component scores for axes 1-2...". Please also check there are no other occurrences of "principle". Figure 7 – "...and discharge (c-f)" References: Carlisle et al. 2017; Fuhrman et al. 2018; Isaak et al. 2016b; Josse and Husson 2012; Steel et al. 2017 not cited in text. Inconsistencies in citations: Line 51 – Rieman et al 2015a; Line 80 Piechota 2001 or 1997?; Line 84 Gallacher 2016 or 2017?; line 90 Trumbo et al. Not referenced; line 175 – correct to R Development Core Team; Line 205 correct to SAS Institute Inc.; line 326 – spelling of Nusslé; line 352 – Jackson et al. 2017 or 2018?

Table 3 not cited in text.

---

## Referee Comment (RC2) · N. Rivers-Moore (Referee) · 10 Aug 2018

**Hess – Isaak et al.: Principal components of thermal regimes in mountain river networks**

**General comments**

This manuscript presents an elegant analysis of different components (magnitude, frequency, duration, timing) of thermal events for a large number of time series points in the United states. Two variations of Principal Components Analysis (T-mode and S-mode analyses) refine the analysis very nicely into spatial regions and temporal seasons of thermal homogeneity and seasonality. By disaggregating time series into metrics, and accounting for high levels of redundancy between metrics, together with the PC analyses, this research presents a novel approach to optimising site locations for water temperature gauging networks. In my opinion, this is a very useful addition to thermal research in lotic systems. The approach is generic and applicable to a global audience.

The manuscript is clearly and well written, methodologically elegant and scientifically sound. I recommend publication given minor comments corrected below.

**Specific comments**

Section 3.1 – Does one need to specify that the study assumed stationarity in the data, in order to generate temperatures for 365 days based on five-year time series?

Line 207 – Please provide a summary of the environmental gradients; it may be worth including a table on these.

Line 212 – Please explain why the thermal year started on 1 December. In South Africa, we typically use 1 October – 30 September for the Hydrological year, but I am aware that this varies regionally, being based on the onset of the highest discharge season.

Line 234 – It would make more sense to me to represent the thermal gradient per 100m. This would be a useful figure in defining a water temperature lapse rate. For air temperatures, this is typically expressed as something like 0.7°C per 100m.

**Technical comments**

Title and elsewhere in text: please check for correct spelling of "Principle [as in components]", which needs to be corrected to PRINCIPAL and checked throughout text, as there are instances of both. Nothing serious – I get confused between these two spellings!

Lines 37-40: Sentence does not read well. Suggested revision "Knowledge of the local thermal regime, based on the annual sequence of temperatures characteristic to specific locations within a river network, is key to understanding natural conditions and diagnosing anthropogenic impairments."

Lines 62-64: Suggested revision "While that may bring..., most warm stream...correlated with each other and therefore redundant. If redundancy is also reflected across a broader..."

Paragraph beginning line 146: Be explicit that these time series refer to water temperatures, as later on in the manuscript air temperatures are also used.

Line 218 – "sites, an S-mode"

Line 253 – Figure 4a

Line 257 – insert Figure 4c

Table 1 – write US in full; standardise on number of decimal points down columns (also applies for Table 3).

Figure 2 – I like this figure! Please include the range of $R^2$ values, and I would recommend that the caption explicitly describes the month(s) with the highest thermal gradient.

Figure 4 – caption revision to say "...show principal component scores for axes 1-2...". Please also check there are no other occurrences of "principle".

Figure 7 – "...and discharge (c-f)"

References: Carlisle et al. 2017; Fuhrman et al. 2018; Isaak et al. 2016b; Josse and Husson 2012; Steel et al. 2017 not cited in text.

Inconsistencies in citations: Line 51 – Rieman et al 2015a; Line 80 Piechota 2001 or 1997?; Line 84 Gallacher 2016 or 2017?; line 90 Trumbo et al. Not referenced; line 175 – correct to R Development Core Team; Line 205 correct to SAS Institute Inc.; line 326 – spelling of Nusslé; line 352 – Jackson et al. 2017 or 2018?

Table 3 not cited in text.

---

## Referee Comment (RC3) · Anonymous Referee #2 · 28 Aug 2018

This manuscript provides a nice analysis, characterizing the spatial and temporal characteristics and controls of thermal regimes of stream water. The work is based on a novel application of Principal Component Analysis, including the highly interesting differentiation of T-mode and S-mode PCA to illustrate both, temporal and spatial consistency of the stream temperature pattern.

The paper is very well and concisely written, including a clear and complete description of the data and methods used.

However and despite the flawless implementation of the analysis, the interpretation of the results and their implications remain somewhat superficial. After reading the manuscript, it seemed to me that the authors contented themselves with demonstrating how a well-known statistical tool can be applied with stream temperature data. The

one finding that I found most interesting to demonstrate the value of PCA was that the authors could pin down the timing of the phase transitions.

I may not see the forest for the trees but apart from that I am not sure what can be learned from the analysis. As far as I understand, the results essentially suggest that (1) stream temperature is mostly controlled by temperature magnitudes and lengths of winter periods (which again is related to temperature magnitude one would assume) and (2) stream temperature is more spatially homogeneous in winter than in summer. While the first does not really come as a surprise, it seems that the latter can also be inferred without PCA (or in other words: how is the information content of Figure 2 different to that of Figure 6?).

I would thus be glad if the authors could invest a bit more effort in (1) highlighting the benefits of PCA with respect to other methods and (2) providing a somewhat stronger synthesis of their results – what are the novel aspects that can be learned from these results?

Technical comments:

p.7,l.204: what is a "Princomp procedure"?

p.7,l.212: is there a specific reason to run the T-mode PCA on the 5-year mean values of the daily mean temperatures? In other words, why use 365 days (i.e. columns) and not the full data set of 1826 as in the S-mode analysis?

p.18,table 1: the values for reach slope seem excessively small. Should the unit perhaps be [m/m]? Please check.
* * *

---

## Author Comment (AC1) · 12 Sep 2018

General comments This manuscript presents an elegant analysis of different components (magnitude, frequency, duration, timing) of thermal events for a large number of time series points in the United states. Two variations of Principal Components Analysis (T-mode and S-mode analyses) refine the analysis very nicely into spatial regions and temporal seasons of thermal homogeneity and seasonality. By disaggregating time series into metrics, and accounting for high levels of redundancy between metrics, together with the PC analyses, this research presents a novel approach to optimising site locations for water temperature gauging networks. In my opinion, this is a very useful addition to thermal research in lotic systems. The approach is generic and applicable to a global audience. The manuscript is clearly and well written, methodologically elegant

and scientifically sound. I recommend publication given minor comments corrected below.

Our response: We much appreciate the reviewer's kind words and attention to detail in their comments. We are willing to make many of the suggested revisions as described below.

Specific comments Section 3.1 – Does one need to specify that the study assumed stationarity in the data, in order to generate temperatures for 365 days based on five-year time series?

Our response: There has been some focus in the recent literature on the possibility of nonstationary responses in stream temperatures due to climate forcing. However, that type of nonstationarity is generally expected over multi-decadal timespans and the prediction is based largely on mechanistic models rather than documentation from empirical trends in monitoring datasets. During the short five year study period we considered, nonstationarity was unlikely to be important and the 12% of missing daily observations were reconstructed from nearby sites with strong covariance using the missMDA statistical package in R.

Line 207 – Please provide a summary of the environmental gradients; it may be worth including a table on these.

Our response: Table 1 would be expanded to include summaries of these gradients, which were elevation, drainage area, annual flow, and reach slope.

Line 212 – Please explain why the thermal year started on 1 December. In South Africa, we typically use 1 October – 30 September for the Hydrological year, but I am aware that this varies regionally, being based on the onset of the highest discharge season.

Our response: The water year in North America is also considered to start October 1 but in the climatological literature, seasons are slightly different and considered to

be winter (Dec/Jan/Feb), spring (March/April/May), summer (June/July/Aug), and fall (Sept/Oct/Nov). Climatological considerations seemed more relevant in the present context, and the conventional three month seasonal periods also conveniently matched the water temperature patterns common to our study site. We describe our rationale for starting the thermal year on Dec 1 in the manuscript where we state "We started the thermal year on December 1 because temperatures usually reach their annual lows by this date and the 3-month period thereafter constituted a logical winter season (i.e., December, January, February)."

Line 234 – It would make more sense to me to represent the thermal gradient per 100m. This would be a useful figure in defining a water temperature lapse rate. For air temperatures, this is typically expressed as something like 0.7°C per 100m.

Our response: We agree and are willing to revise the values in Figure 2 accordingly.

Technical comments Title and elsewhere in text: please check for correct spelling of "Principle [as in components]", which needs to be corrected to PRINCIPAL and checked throughout text, as there are instances of both. Nothing serious – I get confused between these two spellings!

Our response: Embarrassing on our part that we missed this. Inconsistent usage will be changed to "principal" throughout.

Lines 37-40: Sentence does not read well. Suggested revision "Knowledge of the local thermal regime, based on the annual sequence of temperatures characteristic to specific locations within a river network, is key to understanding natural conditions and diagnosing anthropogenic impairments."

Our response: This revision would be made.

Lines 62-64: Suggested revision "While that may bring..., most warm stream...correlated with each other and therefore redundant. If redundancy is also reflected across a broader..."

[Figure]

Our response: This revision would be made.

Paragraph beginning line 146: Be explicit that these time series refer to water temperatures, as later on in the manuscript air temperatures are also used.

Our response: This revision would be made.

Line 218 – "sites, an S-mode"

Our response: This change would be made.

Line 253 – Figure 4a

Our response: This change would be made.

Line 257 – insert Figure 4c

Our response: This change would be made.

Table 1 – write US in full; standardise on number of decimal points down columns (also applies for Table 3).

Our response: United States would be spelled out. In the tables, we generally standardized on having two or three significant digits rather than the number of decimal points. We are glad to adjust this either way depending on the convention in HESS but have left the values unchanged for now.

Figure 2 – I like this figure! Please include the range of R2 values, and I would recommend that the caption explicitly describes the month(s) with the highest thermal gradient.

Our response: We are willing to add the R2 values to the figure but probably would not modify the caption to highlight a subset of months with the highest thermal gradients because we don't think that information is inherently more useful than that for other months.

Figure 4 – caption revision to say "...show principal component scores for axes 1-2...".

Please also check there are no other occurrences of "principle".

Our response: These revisions would be made.

Figure 7 – "...and discharge (c-f)"

Our response: Caption would be revised accordingly.

References: Carlisle et al. 2017; Fuhrman et al. 2018; Isaak et al. 2016b; Josse and Husson 2012; Steel et al. 2017 not cited in text.

Our response: These errors would be corrected.

Inconsistencies in citations: Line 51 – Rieman et al 2015a; Line 80 Piechota 2001 or 1997?; Line 84 Gallacher 2016 or 2017?; line 90 Trumbo et al. Not referenced; line 175 – correct to R Development Core Team; Line 205 correct to SAS Institute Inc.; line 326 – spelling of Nusslé; line 352 – Jackson et al. 2017 or 2018?

Our response: These inconsistencies would be rectified.

Table 3 not cited in text.

Our response: An appropriate citation would be added.

---

## Author Response (AR1)

Dear Dr. Freer,

Please find our reviewer responses and descriptions of revisions to HESS-2018-266 below. In addition to the requested revisions, we have also improved the manuscript with these additional revisions: 1) revised all figures for improved clarity, 2) added one paragraph of new results that highlights additional nuances regarding stream temperature dynamics and associated basin properties, 3) fully revised and expanded the discussion section. We hope you find the revision and responses to reviewer comments satisfactory and look forward to future correspondence regarding this manuscript. Best regards, Dan Isaak

**Reviewer #1 comments**
**General comments**
This manuscript presents an elegant analysis of different components (magnitude, frequency, duration, timing) of thermal events for a large number of time series points in the United states. Two variations of Principal Components Analysis (T-mode and S-mode analyses) refine the analysis very nicely into spatial regions and temporal seasons of thermal homogeneity and seasonality. By disaggregating time series into metrics, and accounting for high levels of redundancy between metrics, together with the PC analyses, this research presents a novel approach to optimising site locations for water temperature gauging networks. In my opinion, this is a very useful addition to thermal research in lotic systems. The approach is generic and applicable to a global audience. The manuscript is clearly and well written, methodologically elegant and scientifically sound. I recommend publication given minor comments corrected below.

Our response: We much appreciate the reviewer's kind words and attention to detail in their comments. We have made many of the suggested revisions as described below.

Specific comments
Section 3.1 – Does one need to specify that the study assumed stationarity in the data, in order to generate temperatures for 365 days based on five-year time series?

Our response: There has been some focus in the recent literature on the possibility of nonstationary responses in stream temperatures due to climate forcing. However, that type of nonstationarity is generally expected over multi-decadal timespans and the prediction is based largely on mechanistic models rather than documentation from empirical trends in monitoring datasets. During the short five year study period we considered, nonstationarity was unlikely to be important and the 12% of missing daily observations were reconstructed from nearby sites with strong covariance using the missMDA statistical package in R.

Line 207 – Please provide a summary of the environmental gradients; it may be worth including a table on these.

Our response: Table 1 was expanded to include summaries of these gradients, which were elevation, drainage area, annual flow, and reach slope.

Line 212 – Please explain why the thermal year started on 1 December. In South Africa, we typically use 1 October – 30 September for the Hydrological year, but I am aware that this varies regionally, being based on the onset of the highest discharge season.

Our response: The water year in North America is also considered to start October 1 but in the climatological literature, seasons are slightly different and considered to be winter (Dec/Jan/Feb), spring (March/April/May), summer (June/July/Aug), and fall (Sept/Oct/Nov). Climatological considerations seemed more relevant in the present context, and the conventional three month seasonal periods also conveniently matched the water temperature patterns common to our study site. We describe our rationale for starting the thermal year on Dec 1 in the manuscript where we state "We started the thermal year on December 1 because temperatures usually reach their annual lows by this date and the 3-month period thereafter constituted a logical winter season (i.e., December, January, February)."

Line 234 – It would make more sense to me to represent the thermal gradient per 100m. This would be a useful figure in defining a water temperature lapse rate. For air temperatures, this is typically expressed as something like 0.7°C per 100m.

Our response: We agree and revised the values in Figure 2 accordingly.

Technical comments
Title and elsewhere in text: please check for correct spelling of "Principle [as in components]", which needs to be corrected to PRINCIPAL and checked throughout text, as there are instances of both. Nothing serious – I get confused between these two spellings!

Our response: Embarrassing on our part that we missed this. Inconsistent usage was changed to "principal" throughout.

Lines 37-40: Sentence does not read well. Suggested revision "Knowledge of the local thermal regime, based on the annual sequence of temperatures characteristic to specific locations within a river network, is key to understanding natural conditions and diagnosing anthropogenic impairments."

Our response: This revision was made.

Lines 62-64: Suggested revision "While that may bring..., most warm stream...correlated with each other and therefore redundant. If redundancy is also reflected across a broader..."

Our response: This revision was made.

Paragraph beginning line 146: Be explicit that these time series refer to water temperatures, as later on in the manuscript air temperatures are also used.

Our response: This revision was made.

Line 218 – "sites, an S-mode"

Our response: This change was made.

Line 253 – Figure 4a

Our response: This change was made.

Line 257 – insert Figure 4c

Our response: This change was made.

Table 1 – write US in full; standardise on number of decimal points down columns (also applies for Table 3).

Our response: United States was spelled out. In the tables, we generally standardized on having two or three significant digits rather than the number of decimal points. We are glad to adjust this either way depending on the convention in HESS but have left the values unchanged for now.

Figure 2 – I like this figure! Please include the range of R2 values, and I would recommend that the caption explicitly describes the month(s) with the highest thermal gradient.

Our response: We added the R2 values to the figure but did not modify the caption to highlight a subset of months with the highest thermal gradients because we don't think that information is inherently more useful than that for other months.

Figure 4 – caption revision to say "...show principal component scores for axes 1-2...". Please also check there are no other occurrences of "principle".

Our response: These revisions were made.

Figure 7 – "...and discharge (c-f)"

Our response: Caption was revised accordingly.

References: Carlisle et al. 2017; Fuhrman et al. 2018; Isaak et al. 2016b; Josse and Husson 2012; Steel et al. 2017 not cited in text.

Our response: These errors were corrected.

Inconsistencies in citations: Line 51 – Rieman et al 2015a; Line 80 Piechota 2001 or 1997?; Line 84 Gallacher 2016 or 2017?; line 90 Trumbo et al. Not referenced; line 175 – correct to R Development Core Team; Line 205 correct to SAS Institute Inc.; line 326 – spelling of Nusslé; line 352 – Jackson et al. 2017 or 2018?

Our response: These inconsistencies were rectified.

Table 3 not cited in text.

Our response: An appropriate citation was added.

**Reviewer #2 comment:**
**Anonymous Referee #2**

This manuscript provides a nice analysis, characterizing the spatial and temporal characteristics and controls of thermal regimes of stream water. The work is based on a novel application of Principal Component Analysis, including the highly interesting differentiation of T-mode and S-mode PCA to illustrate both, temporal and spatial consistency of the stream temperature pattern. The paper is very well and concisely written, including a clear and complete description of the data and methods used. However and despite the flawless implementation of the analysis, the interpretation of the results and their implications remain somewhat superficial. After reading the manuscript, it seemed to me that the authors contented themselves with demonstrating how a well-known statistical tool can be applied with stream temperature data. The one finding that I found most interesting to demonstrate the value of PCA was that the authors could pin down the timing of the phase transitions. I may not see the forest for the trees but apart from that I am not sure what can be learned from the analysis. As far as I understand, the results essentially suggest that (1) stream temperature is mostly controlled by temperature magnitudes and lengths of winter periods (which again is related to temperature magnitude one would assume) and (2) stream temperature is more spatially homogeneous in winter than in summer. While the first does not really come as a surprise, it seems that the latter can also be inferred without PCA (or in other words: how is the information content of Figure 2 different to that of Figure 6?). I would thus be glad if the authors could invest a bit more effort in (1) highlighting the benefits of PCA with respect to other methods and (2) providing a somewhat stronger synthesis of their results – what are the novel aspects that can be learned from these results?

Our response: We agree with the overall critique that greater interpretation of results would be beneficial so have revised and expanded the discussion in a subsequent revision. As for the reviewer's first comment that "(1) stream temperature is mostly controlled by temperature magnitudes and lengths of winter periods (which again is related to temperature magnitude one would assume)", the statement in the parenthetical clause is incorrect in conflating temperature magnitude with the length of the winter period. Our analysis reveals that these are instead two distinct aspects of thermal regimes in the mountain streams we studied. Streams with similar mean or maximum summer temperatures appear to vary considerably with regards to their winter period lengths when temperatures are largely homothermous. Exploring why that variation occurs was a useful addition to a discussion revision. The reviewer's latter point that "(2) stream temperature is more spatially homogeneous in winter than in summer. […], it seems that the latter can also be inferred without PCA (or in other words: how is the information content of Figure 2 different to that of Figure 6?)" is accurate but had previously been documented only at a few sites using time series plots like Figure 2. The T-mode PCA results put that site-level pattern into a broader context composed of hundreds of sites across large river basins. In this particular dataset, the thermal pattern across all the sites during the winter was largely consistent but that consistency was unknown prior to the analysis. Moreover, it is unlikely to be repeated in subsequent analyses we are planning with larger datasets that encompass greater climatic and hydrological diversity, so these PCA tools may help us identify subdomains regionally wherein stream thermal regimes behave differently. Here again, we think the revised discussion section has done much to bring out these points.

Technical comments:
p.7,l.204: what is a "Princomp procedure"?

Our response: This was the statistical script run in the SAS software to perform the analysis. The reference to SAS was moved forward in this sentence so it is adjacent to the "Princomp procedure" reference for clarity.

p.7,l.212: is there a specific reason to run the T-mode PCA on the 5-year mean values of the daily mean temperatures? In other words, why use 365 days (i.e. columns) and not the full data set of 1826 as in the S-mode analysis?

Our response: We judged it unlikely that appreciable inter-annual differences would be observed in the spatial phases revealed by the T-mode analysis given the large elevational gradient in the study area and because the dominant patterns in PC loadings were driven by cold and warm season cycles (Figure 5). Showing one annual cycle of tradeoffs between PC1 and PC2 was easier to present and read, so we elected to run the analysis on the 5-year mean daily values. We were less certain regarding the potential consistency of inter-annual variation in temporal patterns described using the S-mode analysis, so ran that analysis on the disaggregated water temperature records as well. In retrospect, the results based on the disaggregated records yielded similar insights as those based on the 5-year mean dailies, so little new information was gained except to re-enforce the fact that water temperatures respond strongly to variation in air temperature and discharge across a range of climate year conditions.

other than being repeated 4 more times in the plot of loadings. Displaying the pattern over the course the dominant annual scale variability should be more informative for readers and more easily grasped.

p.18,table 1: the values for reach slope seem excessively small. Should the unit perhaps be [m/m]? Please check.

Our response: Yes, the units were in m/m rather than % and the label was changed accordingly.

[revised manuscript text omitted]

**Moved up [3]:** Two distinct T-mode spatial phases as during 2-6 months of year, homothermous and no response but outside of this time, S-mode revealed strong response to air temperatures. Discharge effect was much smaller but showed noteworthy mediation and hysteresis wherein sensitivity to air temperature variation was muted on the ascending limb as others have noted (Mohseni 98) and enhance on descending limb

**Moved up [2]:** Would fit within Maheu et al. (2015) scheme as cold stable/cold unstable that are characterized by low magnitude and winter homothermous period.

**Moved (insertion) [2]**

[revised manuscript text omitted]